# Effectiveness of Exercise Loading on Bone Mineral Density and Quality of Life Among People Diagnosed with Osteoporosis, Osteopenia, and at Risk of Osteoporosis—A Systematic Review and Meta-Analysis

**DOI:** 10.3390/jcm14124109

**Published:** 2025-06-10

**Authors:** Saeed Mufleh Alnasser, Reem Abdullah Babakair, Amal Fahad Al Mukhlid, Salihah Saleh Saeed Al hassan, Shibili Nuhmani, Qassim Muaidi

**Affiliations:** 1Alyvia Rehabilitation Center, Abha Private International Hospital, Abha 1794, Saudi Arabia; saeed.alnasser@aph.med.sa (S.M.A.); amal.almukhlid@aph.med.sa (A.F.A.M.); sah20201419@hotmail.com (S.S.S.A.h.); 2Department of Physical Therapy, College of Applied Medical Sciences, Imam Abdulrahman Bin Faisal University, Dammam 31451, Saudi Arabia; babkairra@gmail.com (R.A.B.); qmuaidi@iau.edu.sa (Q.M.)

**Keywords:** osteoporosis, exercise therapy, bone density, resistance training, quality of life

## Abstract

**Background:** This systematic review and meta-analysis aims to provide a detailed analysis of the current state of knowledge on Progressive Exercise Training (PET), encompassing its diverse modalities, effects on bone mineral density (BMD), quality of life outcomes, and implications for clinical practice. **Methods:** A structured search strategy was employed to retrieve literature from seven databases (PubMed, Web of Science, Scopus, MEDLINE, Science Direct, EBSCO, CINHAL, and PEDro) yielded twenty-four randomized controlled trials (RCTs) meeting the inclusion criteria. The methodological quality of studies was evaluated using the PEDro scale. Meta-analyses were carried out to comprehensively assess the collective impact of PET on bone mineral density outcomes. **Results:** PET exhibited favorable effects on BMD across multiple anatomical sites, encompassing the femoral neck, total hip, lumbar spine, and others. This effect was observed across different age groups and genders, highlighting its potential benefits for diverse populations. PET encompasses a range of modalities, including resistance training, aerobic training, impact training, whole-body vibration, and tai chi, with a duration ranging from 4 to 24 months, with weekly sessions varying from two to five times. Some studies combined these modalities, reflecting the adaptability of PET to individual preferences and capabilities. Tailoring exercise prescriptions to individual needs emerged as a feasible approach within PET. A subset of studies assessed quality of life using validated instruments such as the 36-item short form survey (SF-36), shortened osteoporosis quality of life questionnaire (SOQLQ), and menopause quality of life instrument (MENQOL). **Conclusions:** This study provides strong evidence that PET represents a promising intervention for osteoporosis management, enhancing BMD and, to some extent, quality of life. PET offers a beacon of hope for better skeletal health and well-being in individuals grappling with osteoporosis, emphasizing the need for its incorporation into clinical practice.

## 1. Introduction

Osteoporosis is an important public health issue worldwide. It is a progressive disease characterized by decreased bone mineral density [1]. Osteoporosis affects 200 million women globally, affecting 10% of those under 60 years, 20% between 60 and 70 years, 40% between 80 and 90 years, and 67% over 90 years [2]. An estimated 8.9 million osteoporotic fractures occur globally each year, with one fracture taking place every three seconds [3]. Compared to one in five men with osteoporosis, one in three women experiences a fracture [3]. Due to the growing number of elderly people, osteoporosis is becoming a significant public health concern [4,5]. The incidence of osteoporosis has been calculated to be 4950/1,000,000 person-years globally [6]. In osteoporosis, the microarchitecture of the bone thins and becomes porous, making the bones more brittle and raising the risk of fracture. As people get older, these changes become more frequent, especially in postmenopausal women, because of a rise in osteoclastic activity and a decrease in osteoblastic activity [7]. Individuals with osteoporosis often become involved with low-impact fractures because of their brittle bones [8]. These fractures can range in severity from providing only minor discomfort to losing functional independence and possibly death [8]. For example, hip fractures are the most serious consequence of osteoporosis, which is the leading cause of death and morbidity and results in the most significant financial burden [9]. A systematic review of 130 studies and more than 670,000 hip fracture patients has estimated the medical and social costs to be USD 43,669 per patient every year [10].

Bone is a dynamic tissue that adjusts its mass, structure, and/or strength in response to changes in mechanical stresses so that it can sustain upcoming demands and avoid fracture. This response is regulated by a negative feedback system [11]. The key characteristics include dynamic intermittent loading rather than static high magnitude and fast loading applied in different or unusual directions or patterns, and few loading repetitions, during adequate loading conditions. The bone’s ability to respond to continuous stress decreases with time or with increasing repetitions because bone cells become desensitized to recurrent stimulation. For instance, there is proof that performing short bursts of loading followed by rest intervals promotes osteogenesis more than constantly performing the same number of loads [12]. These findings are significantly taken as a whole since they helped shape the creation of clinical exercise prescription guidelines for the prevention and treatment of osteoporosis [11]. The American College of Sports Medicine (ACSM) has also recommended considering different training concepts to increase training adaptations when creating any exercise program to improve bone health [13]. In loading programs, the overload principle is crucial. It is necessary to gradually increase the loading stimulus as bone adapts. Loads applied to the bone by gravitational or muscular forces have to be greater than the typical loads experienced in daily activities. This notion is supported by Frost’s mechanistic theory, which states that bones have a threshold level of adaptation known as the Minimum Effective Strain (MES). This theory states that stresses above or below this threshold will promote either bone production or resorption, respectively, resulting in an improvement or decline in bone strength. While the volume of stress delivered to the bone is a key component of overload training, additional aspects to consider when designing an exercise program to support bone health include loading pattern, rate, number, and frequency of exercise. The treatment of postmenopausal osteoporosis can involve any exercise intensity. Weight-bearing and non-weight-bearing activities, in addition to pharmacological treatment, can considerably improve bone mineral density and quality of life in elderly adults with osteoporosis [14]. The quality of life, balance, and strength of osteoporosis patients are significantly improved by physical therapy [15]. By increasing BMD over the course, aerobic dance therapy helped the osteoporosis population to improve their quality of life and lower their risk of bone fracture [15]. Commonly, exercises prescribed to promote bone remodeling are categorized as either impact, low impact, or non-impact exercises, through various modes such as strengthening, stretching, aerobic, or combinations thereof. High-impact exercise regimes which impose high strains with different modes have proven to be an ideal exercise tool to promote unaccustomed stress on the bone which enhances bone remodeling. Jumping, which provides a short burst of force, was the common mode of exercise used to provide a high impact on musculoskeletal structures, which showed significant improvement and maintenance of BMD, muscle strength, coordination, and components of balance [16]. Previous meta-analysis on both premenopausal and postmenopausal women inferred that high-impact exercise protocols profoundly improved BMD, muscle strength, and balance [17]. High-impact exercises that provide various modes, intensities, frequencies, and duration showed convincing improvements in premenopausal but not in postmenopausal women [18]. It was found that a shift in the center of mass of the femoral neck towards the medial, a raise in the threshold for osteogenesis, a decrease in the muscle mass, and an increase in the latency response are some of the factors for poor or no improvement among the risk factors associated with osteoporosis [19]. Research on premenopausal women using high-impact exercises with various modes of intensity, frequency, and repetitions has concluded that the exercise regime improved the BMD in the femoral neck but not in the greater trochanter and vertebrae [20]. However, a progressive high-intensity and -frequency exercise protocol has shown a substantial enhancement in BMD values and positive effects on other osteoporotic risk factors in premenopausal women [21]. This literature review intends to explore whether progressive modification in the intensity, repetition, frequency, and duration has a prominent effect on BMD and quality of life.

The literature identifies exercise training as the only way to reduce all modifiable risk factors of fracture, including fall risk, fall impact, and bone strength. Progressive loading exercises play an important role in improving bone mineral density and quality of life in osteoporotic patients. The literature shows a positive effect of exercises such as resistance, aerobics, impact, whole-body vibration, and a combination of exercises on BMD and quality of life among people diagnosed with osteoporosis or osteopenia. However, therapists still determine an appropriate exercise for osteoporotic or osteopenia subjects. To obtain a precise determination of the best exercise, we have formulated the following research questions: What is the effect of different exercise loadings on bone mineral density and quality of life among individuals diagnosed with osteoporosis or osteopenia, or at risk of osteoporosis? And which exercise is more beneficial for these study populations? Hence, the objective of the present systematic review and meta-analysis is to delineate the effect sizes of different exercises and identify which exercise has superior effects in improving BMD and quality of life among people diagnosed with osteoporosis or osteopenia, or at risk of osteoporosis.

## 2. Materials and Methods

The current study is registered in the OSF registries platform with the following registration DOI: https://doi.org/10.17605/OSF.IO/PSH3F. The review was conducted following the PRISMA 2020 framework to ensure methodological rigor and clear reporting of the review processes.

### 2.1. Eligibility Criteria

This systematic review and meta-analysis considered randomized controlled trials (RCTs) that applied progressive loading exercises. Studies that measured and analyzed both BMD and QOL or either of these outcomes were eligible for inclusion. Studies published in languages other than English were excluded from this review. These studies involved participants with no diagnosis of osteoporosis, osteopenia, and risk of osteoporosis, single-session studies, research protocols, and studies with experimental groups that did not adhere to the specified exercise types.

### 2.2. Search Strategy

Two independent reviewers (S.M.A. and S.N.) conducted a comprehensive search for English-language articles published up to 2022. Initially, one reviewer screened titles and references for relevance, followed by further screening based on the defined selection criteria.

Differences in the selection of studies were addressed through the intervention of an experienced reviewer. Our comprehensive search, designed to encompass a wide range of pertinent studies in our review including PubMed, Web of Science, Scopus, MEDLINE, Science Direct, EBSCO, CINAHL, and PEDro. To optimize our search, we employed a strategy that incorporated MeSH keywords, followed the PICO format, and judiciously utilized Boolean operators (AND, OR). Furthermore, we expanded our pool of relevant articles by meticulously reviewing the reference lists of the articles we had initially retrieved (Table 1).

### 2.3. Data Collection Process

Data extraction from the chosen studies was independently performed by two reviewers (S.M.A and S.N). In instances of disparities, a third reviewer was engaged to reconcile any differences in opinions. Information was systematically collected across six distinct categories, including patient demographics, the PEDro score, sample size, eligibility criteria, outcome measures, and the specific exercise regimens prescribed for both the experimental and control groups. Additionally, we considered any tools or methods used for monitoring progression throughout the studies.

### 2.4. Assessment of Methodological Quality and Grading of Evidence

In our assessment of the methodological quality of the included randomized controlled trials (RCTs), we utilized the PEDro scale. This scale assigns scores, with ratings falling within the range of 9–10 indicating “excellent”, 6–8 indicating “good”, 4–5 indicating “fair”, and scores below 4 denoting “poor” methodological quality. To ensure the rigor of this evaluation, two independent assessors conducted the assessment, and any disparities were resolved through the intervention of a third assessor.

Additionally, RCTs deemed of high quality, with a PEDro score of at least 6, were further classified into two subcategories: level 1a, denoting studies with more than one instance of a PEDro score of 6, and level 1b, indicating studies with a single instance of a PEDro score of 6. RCTs of poor quality, characterized by a PEDro score below 6, were classified as level 2 evidence. This classification approach was adapted from the framework proposed by Sackett et al., based on the number of high-quality trials supporting each specific outcome measure.

### 2.5. Assessment of Risk of Bias

To assess bias within the chosen studies, we employed the Review Manager (RevMan) software version 5.4.1 (Cochrane Collaboration, Oxford, UK). Two separate reviewers conducted assessments across various domains, which encompassed allocation concealment, random sequence generation, blinding of participants, blinding of outcome data, handling of incomplete outcome data, selective reporting, disparities in interventions between groups, and other potential sources of bias. The categorizations used to describe each domain were “high risk”, “low risk”, or “ambiguous”. In instances where discrepancies emerged between the two reviewers, a third expert reviewer was consulted to determine the appropriate categorizations.

### 2.6. Data Synthesis

To facilitate data synthesis, effect sizes for QOL and BMD outcomes were calculated using meta-analysis when these outcomes were reported in two or more studies. In cases where descriptive values were presented as medians and ranges, means and standard deviations were derived using established conversion formulas. Any missing information was diligently acquired by contacting the respective authors via email. The meta-analysis was conducted employing the RevMan software version 5.4.1 (Cochrane Collaboration, Oxford, UK).

Heterogeneity analysis, as indicated by I2 statistics, was performed to assess clinical heterogeneity among the characteristics of the included studies. In cases where no significant clinical heterogeneity was identified, a random-effects model was utilized for analysis. Pooled standardized mean differences (SMDs) were computed for the outcome measures. Furthermore, subgroup analyses were conducted based on the type of exercise (resistance, impact, aerobic, flexibility, or balance training, either individually or in combination) and progressive loading factors (intensity, duration, frequency, or repetitions).

## 3. Results

### 3.1. Search Outcomes

A thorough search of multiple databases produced 1538 studies. Twenty-four studies were included in the review process following rigorous screening and selection. The detailed procedures for database searches, study exclusion, and inclusion criteria are outlined in Figure 1.

### 3.2. Characteristics of the Included Studies

Across all the twenty-four studies, 1961 subjects took part in the trials, and 1235, 396, 271, and 59 were postmenopausal women (PMPW), elderly women (EW), pre-menopausal women (PRMPW), and middle- and older-aged men diagnosed with osteoporosis or risk of osteoporosis, respectively. The population included in the studies was aged 35–80 years. All the subjects included in the trials were suffering either from ischemic or hemorrhagic strokes. However, considering the participants included among trials, fifteen studies included PMPW [22,23,24,25,26,27,28,29,30,31,32,33,34,35,36]. Five studies included EW [25,37,38,39,40]. Three studies included PRMPW. One study included middle- and older-aged men [41].

All included RCTs compared progressive exercise with standard physical therapy or other exercises or with no exercise or sedentary groups. The mode of exercise given in experimental groups varied among the included studies. In seven studies, the intervention prescribed in the experimental group was resistance training (RT) [27,28,35,36,37,39,40]; in five studies, aerobic training (AT) [21,23,24,34,40]; in five studies, impact training (IT) [21,25,26,42,43]; in two studies, whole body vibration (WBV) [27,44]; in one study, tai chi [22]; and in eight studies, combination exercise training (CET) [29,30,31,32,33,37,38,41]. The total duration of prescription of exercises ranged from 4 months to 24 months, with number of sessions per week from two to five. Table 2 outlines the information on study participants, types of interventions, and corresponding outcome measures. The age range of the participants across the studies was 35–80 years.

Among the included studies, fifteen focused on PMPW, five included EW, three incorporated PRMPW, and one involved middle-aged and older men. Each of the included RCTs compared progressive exercise interventions with conventional physiotherapy, other exercise modalities, or no exercise (sedentary) groups. The specific type of exercise prescribed in the experimental groups varied across studies and included resistance training (RT) in seven studies, aerobic training (AT) in five studies, impact training (IT) in five studies, whole-body vibration (WBV) in two studies, tai chi in one study, and combination exercise training (CET) in eight studies. The duration of exercise prescriptions ranged from 4 to 24 months, with a weekly session frequency varying from two to five times per week.

### 3.3. Outcome Measures

In 22 experiments, BMD was observed and compared between groups; in 21 trials, BMD in the experimental group either improved significantly or exhibited comparable improvements to that of the control group. Dual-energy X-ray absorptiometry was the equipment predominantly used to evaluate BMD. The sites of bone observed for BMD were Femoral Neck [21,22,24,25,26,28,29,31,32,35,36,39,40,41,43,44], Total Hip [22,27,28,34,38,39,40,41,42], Lumbar Spine [21,22,23,24,28,29,31,32,35,36,39,41,44], Trochanter [21,25,26,28,41,43], Total Temur [21,25,29], Inter Trochanter [21,26,43], Total Body [27,28,34], Proximal Femur [21], Distal Tibia [37], Tibial Shaft [37], Distal Radius [37], Radial Shaft [37], and Calcaneum [37,40] among the included studies, respectively. Quality of life was observed in five studies, and the questionnaires used were 36-item short form surveys (SF 36) [22,33,38], shortened osteoporosis quality of life questionnaire (SOQLQ) [30], and menopause quality of life instrument (MENQOL) [22].

### 3.4. Assessment of Methodological Quality, Level of Evidence, and Bias

Among the included studies, seven studies received a PEDro score of 5 [26,27,30,31,34,35,36], five studies scored 7 [22,28,37,38,40], and four studies each scored 6 [24,25,33,44] and 4 [21,23,42,43].Thirteen studies were classified as fair, and eleven as good, based on the quality assessment. Table 3 presents the PEDro scores assigned to each included study.

According to the results of the risk of bias evaluation, 50% of studies had a high risk of detection bias, and 75% of studies had a high risk of concealed allocation bias. Attrition bias affected 55% of the studies, while participant bias affected all 100% of the studies. The included studies demonstrated a low risk of bias related to outcome reporting and treatment imbalance. A detailed risk of bias assessment is illustrated in Figure 2. Furthermore, our examination of bias provided significant insights into the included studies. Notably, 75% of the studies were found to carry a high risk of concealed allocation, and 50% exhibited a high risk of detection bias. The issue of attrition bias was relevant in 55% of the studies, while all studies were deemed susceptible to participant bias.

Importantly, when it came to reporting bias and treatment imbalance, the studies included in our analysis demonstrated a commendable track record, with low risks identified in these areas. For a comprehensive breakdown of the results from our bias assessment, please consult Figure 2. This comprehensive evaluation contributes to a more thorough understanding of the potential sources of bias within the body of research.

### 3.5. Quantitative Synthesis

A meta-analysis was conducted for BMD at specific sites (femoral neck, total hip, and lumbar spine), given that these were the most frequently assessed bone locations in the included studies. Due to the heterogeneity in the questionnaires used to assess QOL, a meta-analysis for QOL was not performed.

#### 3.5.1. Femoral Neck (BMD) Analysis

When analyzing the effect of progressive exercise training compared to the control group, our findings indicated notably superior outcomes, with a standard mean difference (SMD) of 1.54 at a 95% confidence interval (CI) of 0.76–2.33 (*p* < 0.001).

However, it is essential to acknowledge that there was substantial heterogeneity among the studies, as evidenced by an I^2^ value of 96% (*p* < 0.001). Conversely, the resistance training group did not demonstrate a significant difference when compared to the control groups (SMD = 0.14 at a 95% CI of −1.12–1.40, *p* = 0.82), and this was associated with an I^2^ of 91% (*p* < 0.001). Similarly, the aerobic training group did not exhibit a substantial difference compared to the control groups (SMD = 0.72 at a 95% CI of −1.02–2.46, *p* = 0.42), with an I^2^ of 95% (*p* < 0.001). On the other hand, the whole-body vibration group displayed a significant difference compared to the control groups (SMD = 0.74 at a 95% CI of 0.12–1.36, *p* = 0.02), while the tai chi group exhibited a substantial difference (SMD = 6.90 at a 95% CI of 5.76–8.04, *p* < 0.001). In contrast, the impact training group did not show a significant difference compared to the control groups (SMD = 0.13 at a 95% CI of −0.07–0.33, *p* = 0.19), and there was no significant heterogeneity among these studies (I^2^ = 0% at *p* = 0.72). Lastly, the combined exercise training group demonstrated a significant difference when compared to the control groups (SMD = −6.89 at a 95% CI of −2.27–11.52, *p* = 0.003). However, it is worth noting that this result came with substantial heterogeneity (I^2^ = 98% at *p* < 0.001), indicating variations in study outcomes. For a visual representation of these findings, please refer to Figure 3. This comprehensive analysis contributes valuable insights into the impact of different exercise modalities on femoral neck BMD.

#### 3.5.2. Total Hip (BMD) Analysis

In our examination of the impact of exercise modalities on total hip BMD, progressive exercise training once again displayed notably superior effects when compared to the control group, yielding a standard mean difference (SMD) of 1.34 at a 95% confidence interval (CI) spanning 0.41–2.28 (*p* < 0.001). However, it is worth noting that there was a noticeable level of heterogeneity among the studies, as indicated by an I2 value of 95% (*p* < 0.001).

Conversely, the resistance training group demonstrated a significant difference compared to the control groups (SMD = 0.38 at a 95% CI of 0.03–0.74, *p* = 0.03), with an I2 of 31% (*p* = 0.02). In contrast, the aerobic training group did not exhibit a significant difference when compared to the control groups (SMD = 0.75 at a 95% CI of −0.25–1.75, *p* = 0.14), with an I2 of 82% (*p* = 0.02).

The whole-body vibration group displayed a significant difference when compared to the control groups (SMD = 0.86 at a 95% CI of 0.27–1.45, *p* = 0.004), while the tai chi group exhibited a substantial difference (SMD = 11.34 at a 95% CI of 9.56–13.13, *p* < 0.001).

However, it is important to note that the combined exercise training group did not demonstrate a significant difference when compared to the control groups (SMD = −0.36 at a 95% CI of −0.86–0.13, *p* = 0.15). It is worth mentioning that substantial heterogeneity (I^2^ = 95% at *p* < 0.001) was observed in this particular analysis, indicating variations in study outcomes. For a visual representation of these findings, please consult Figure 4. This comprehensive analysis provides valuable insights into the impact of various exercise interventions on total hip BMD.

#### 3.5.3. Lumbar Spine (BMD) Analysis

Our analysis of lumbar spine BMD revealed significant findings. Progressive exercise training exhibited superior effects when compared to the control group, yielding a standard mean difference (SMD) of 0.95 at a 95% confidence interval (CI) spanning 0.32–1.59 (*p* = 0.003). It is worth noting that the included studies exhibited substantial heterogeneity, reflected by an I^2^ statistic of 94% (*p* < 0.001).

Similarly, the resistance training group also demonstrated a significant difference compared to the control groups (SMD = 0.77 at a 95% CI of 0.42–1.12, *p* < 0.001), and no significant heterogeneity was observed (I^2^ = 0% at *p* = 0.52). In contrast, the aerobic training group did not show a significant difference when compared to the control groups (SMD = 0.37 at a 95% CI of −0.07–0.82, *p* = 0.10), although there was some heterogeneity among the studies at 52% (*p* = 0.12).

Additionally, the whole-body vibration group exhibited a significant difference when compared to the control groups (SMD = 0.76 at a 95% CI of 0.14–1.38, *p* = 0.02). However, both the tai chi group (SMD = −1.04 at 95% CI of −1.49–0.58, *p* < 0.001) and the impact training group (SMD = 0.03 at 95% CI of −0.33–0.39, *p* = 0.87) did not display significant differences when compared to the control groups, with no significant heterogeneity in the latter (I^2^ = 0% at *p* = 0.72).

Lastly, the combined exercise training group demonstrated a significant difference compared to the control groups (SMD = 0.95 at a 95% CI of 0.32–1.59, *p* = 0.003), accompanied by substantial heterogeneity (I^2^ = 94% at *p* < 0.001). For a visual representation of these findings, please consult Figure 5.

Furthermore, when assessing balance in the post-treatment CIMT group, the meta-analysis results did not indicate a significant difference when compared to the control groups. The standardized mean difference (SMD) was 4.94 at a 95% CI spanning −2.48–12.67, with a *p*-value of 0.19, and substantial heterogeneity was observed among the studies (I2 = 92% at *p* < 0.001). In studies that included follow-up assessments, the CIMT group continued to show no significant improvement in balance compared to the control groups (SMD = 3.84 at a 95% CI spanning −2.33–10.01, *p* = 0.22). In this case, heterogeneity among the studies remained notable, with an I2 of 88% at *p* = 0.004. Please refer to Figure 4 for a graphical representation of these results. This comprehensive analysis provides valuable insights into lumbar spine BMD and balance outcomes across various exercise modalities.

## 4. Discussion

Osteoporosis represents a significant public health concern, particularly among postmenopausal women (PMPW), elderly women (EW), pre-menopausal women (PRMPW), and middle-aged and older men at risk of or diagnosed with osteoporosis. In the quest to address this multifaceted challenge, exercise loading has emerged as a promising strategy. This systematic review sought to comprehensively analyze the effects of exercise loading on BMD and quality of life across various demographic groups. The synthesis of evidence from 24 randomized controlled trials (RCTs) not only provides valuable insights into the efficacy of exercise loading but also underscores the methodological intricacies and potential for enhancing osteoporosis management through personalized exercise prescriptions.

### 4.1. The Impact of Exercise Loading Training on BMD

The primary outcome measure in the majority of the included studies was BMD, a key determinant of bone health. Dual-energy X-ray absorptiometry (DXA) was the predominant method employed to evaluate changes in BMD across various anatomical sites, including the femoral neck, total hip, lumbar spine, trochanter, total femur, intertrochanteric, total body, proximal femur, distal tibia, tibial shaft, distal radius, radial shaft, and calcaneum. Notably, the comprehensive assessment of multiple bone sites provides a holistic view of the effects of exercise loading on bone health.

The collective findings from this systematic review suggest that exercise loading has a positive impact on BMD in individuals at risk or diagnosed with osteoporosis. This is evident from the significant improvements in BMD or comparable changes in BMD compared to the control groups. The overarching observation is that exercise loading, as a versatile intervention strategy, holds promise in enhancing bone density across diverse demographic groups. This observation is particularly significant in the context of the aging population, where osteoporosis poses a considerable health burden.

Interestingly, exercise loading beneficial effects on BMD transcend age and gender. Studies included in this review involved participants spanning a wide age range, from 35 to 80 years, indicating the potential utility of exercise loading across the lifespan. The positive outcomes observed in various demographic groups, including PMPW, EW, PRMPW, and middle-aged and older men, underscore the adaptability of exercise loading as an intervention for different populations. Consequently, exercise loading can serve as an inclusive approach to address the evolving demographic landscape and the increasing prevalence of osteoporosis among diverse groups.

### 4.2. Diverse Exercise Modalities in Exercise Loading

One of the notable aspects of this review is the diversity of exercise modalities incorporated into exercise loading interventions. These modalities encompass resistance training (RT), aerobic training (AT), impact training (IT), whole-body vibration (WBV), tai chi, and combination exercise training (CET). Such diversity underscores the flexibility of exercise loading regimens, allowing for tailoring to individual preferences, capabilities, and clinical requirements. This adaptability is a significant advantage, as it ensures that exercise programs are accessible and engaging for a wide range of individuals.

Resistance training, in particular, has demonstrated its effectiveness in improving BMD among participants with osteoporosis. This is consistent with previous research highlighting the importance of mechanical loading in stimulating bone formation. Resistance training programs, often employing low-load, high-repetition strategies, have emerged as a viable means of attenuating BMD losses. These findings provide valuable insights into the utility of resistance training as a cornerstone of PET interventions, especially for individuals who may have limitations or preferences that make other modalities less suitable.

Furthermore, mind–body exercises like tai chi have shown significant benefits in enhancing BMD and postural stability. The inclusion of tai chi in exercise loading regimens highlights the multifaceted nature of these interventions, emphasizing not only physiological but also psychological and emotional well-being. This holistic approach aligns with the broader concept of health promotion and underscores the potential of exercise loading to address the comprehensive needs of individuals with osteoporosis.

Additionally, whole-body vibration (WBV) training and walking have contributed to bone preservation and fall risk reduction. These modalities offer an attractive alternative, particularly for individuals who may have limitations or contraindications for other forms of exercise. The feasibility and effectiveness of WBV and walking programs suggest that PET can be adapted to accommodate diverse clinical scenarios and individual preferences.

### 4.3. The Promise of Multimodal Exercise Loading Programs

Multimodal PET programs, which combine resistance and aerobic exercises, have consistently demonstrated positive effects on BMD and physical performance. Among postmenopausal women and, notably, middle-aged and older men, these programs have showcased the potential for synergistic benefits. The integration of various exercise components into a comprehensive regimen enhances the overall impact on bone health. This emphasizes the concept that exercise loading is not limited to a single modality but can be optimized through thoughtful combinations, potentially delivering superior outcomes.

While the positive outcomes of multimodal exercise loading programs are encouraging, they also highlight the importance of tailoring exercise prescriptions to individual needs and goals. The heterogeneity in participant characteristics, exercise intensity, and duration observed across the studies reinforces the need for personalized approaches. To maximize the benefits of exercise loading, healthcare providers should consider factors such as age, gender, baseline BMD, comorbidities, and exercise preferences when designing individualized programs.

### 4.4. Quality of Life Considerations

Beyond BMD, several studies in this review assessed the impact of exercise loading on the quality of life of individuals with osteoporosis. These investigations utilized validated questionnaires such as the 36-item short form survey (SF-36), shortened osteoporosis quality of life questionnaire (SOQLQ), and menopause quality of life instrument (MENQOL) to gain insights into the broader well-being of participants. While the number of studies directly addressing quality-of-life outcomes remains limited, the findings suggest that PET may have a positive influence on the overall quality of life for individuals with osteoporosis.

Improvements in quality of life encompass not only physical well-being but also psychological and social dimensions. This holistic approach to health underscores the potential comprehensive benefits of exercise loading. As individuals with osteoporosis often face not only physical challenges but also emotional and social implications, interventions that enhance quality of life are of paramount importance.

### 4.5. Methodological Quality and Risk of Bias

An essential component of this systematic review was the assessment of the methodological quality and risk of bias in the included studies. The Physiotherapy Evidence Database (PEDro) scale served as a valuable tool in evaluating the methodological rigor of RCTs. The assessment revealed a spectrum of methodological quality, with scores ranging from 4 to 7, indicating studies of fair to good quality. This variability underscores the need for stringent adherence to methodological standards in future research.

High scores for random allocation, concealed allocation, and baseline comparability were achieved by several studies. However, limitations were observed concerning blinding of participants, therapists, and assessors. Adequate follow-up and intention-to-treat analysis were performed in many studies, enhancing the reliability of the results. The quality assessment process provides valuable insights for both researchers and clinicians, emphasizing the importance of robust study design and execution.

### 4.6. Addressing Sources of Bias

The assessment of the risk of bias demonstrated that 75% of the studies had a high risk of concealed allocation, while 50% showed a high risk of detection bias. Approximately 55% of the studies were susceptible to attrition bias, and all studies were deemed at risk of participant bias. These findings highlight the need for greater attention to mitigating sources of bias in future research. Addressing these limitations will enhance the credibility and generalizability of findings in the field of osteoporosis management through exercise loading.

### 4.7. Limitations

This study has various limitations. First, few studies were found discussing the effect of exercise loading on the quality of life in the study population, which highlights the need for more studies to better understand the interaction between the two variables. Second, there was inadequate control in the random allocation and blinding design of the included studies, which may have increased the risk of bias and, therefore, the reliability of results. This may highlight the need for more studies with better methodological quality, allowing the current systematic review to be repeated in the future. Third, due to the heterogeneity in the types of exercise interventions in the included studies, this might affect the generalizability of the study results. Finally, although subgroup analysis has been conducted in our study based on the type of exercises to find out potential sources of heterogeneity, substantial variability is noticed in some of the pooled outcomes. However, sensitivity analyses or meta regression were not performed due to limited research within subgroups and inconsistent reporting of variables such as exercise intensity, age, and bone mineral density. This restricts our ability to further investigate the source of heterogeneity, and it should be addressed in future studies with more comprehensive data.

### 4.8. Implications and Future Directions

The findings from this systematic review have several implications for clinical practice and future research. Firstly, exercise loading emerges as a versatile and effective approach to osteoporosis management. The observed improvements in BMD, strength, balance, and quality of life underscore the potential of exercise loading as a holistic intervention strategy. Healthcare providers should consider the inclusion of exercise loading as part of a comprehensive management plan for individuals at risk or diagnosed with osteoporosis.

However, to fully realize the benefits of exercise loading, several considerations warrant attention. Firstly, future research should aim to establish specific guidelines regarding exercise prescription, duration, and intensity. While this review demonstrates the efficacy of PET, the optimal strategies for different demographic groups remain to be clarified. Personalized exercise prescriptions that consider age, gender, baseline BMD, and individual preferences are essential to maximizing the benefits of exercise loading.

Moreover, longitudinal studies that assess the long-term effects of exercise loading are needed to provide insights into the sustainability of BMD improvements and the potential for reducing fracture risk. This aspect is particularly crucial in the context of osteoporosis, where long-term management is essential. Furthermore, larger, more diverse populations should be included in future studies to enhance the generalizability of findings. Greater diversity in participant characteristics will enable researchers to explore the potential differential effects of exercise loading among various subpopulations. Addressing methodological limitations identified in the assessed studies is critical. Improving blinding procedures, reducing attrition bias, and enhancing the overall quality of study design and reporting will contribute to more robust evidence in the field. The application of rigorous research standards is essential to further advance our understanding of exercise loading’s role in osteoporosis management. Unfortunately, there is a lack in the literature regarding the effect of progressive exercise loading in patients with osteoporosis; thus, future research should be directed toward examining the effect of progressive exercise intensity and duration.

## 5. Conclusions

This systematic review demonstrates its efficacy in improving BMD across various demographic groups, underscoring its adaptability and potential to address the evolving challenges posed by osteoporosis in aging populations. The diversity of exercise modalities within exercise loading allows for personalized approaches, and the positive outcomes observed in this review emphasize the need to consider PET as an integral component of osteoporosis management. The evidence presented in this systematic review supports the incorporation of PET into osteoporosis management strategies. This versatile approach has the potential to contribute significantly to better skeletal health and quality of life in affected individuals. As the understanding of osteoporosis continues to evolve, PET remains a beacon of hope, offering the promise of improved outcomes and enhanced well-being for those living with this challenging condition.

## Figures and Tables

**Figure 1 jcm-14-04109-f001:**
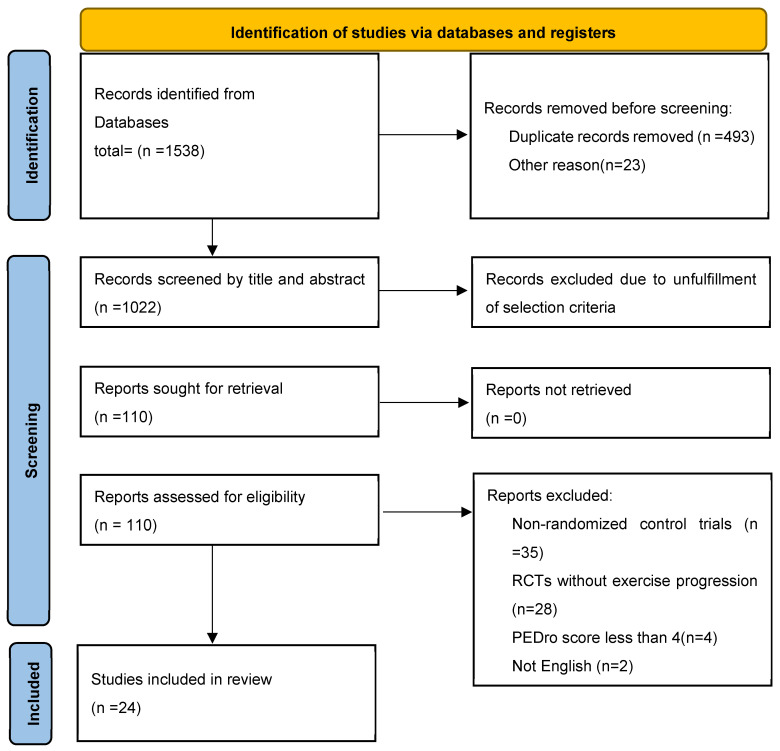
PRISMA flowchart summarizing the identification, screening, eligibility, and inclusion of studies.

**Figure 2 jcm-14-04109-f002:**
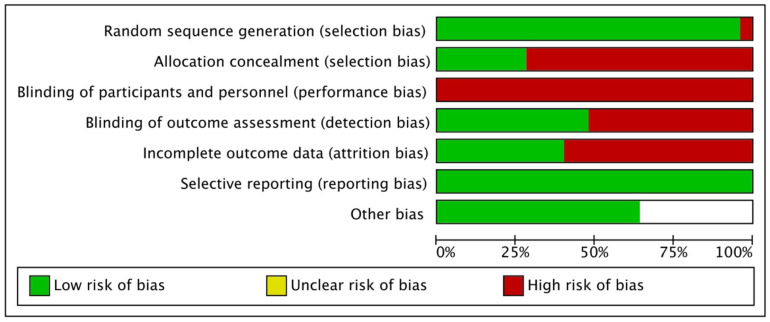
Details of risk of bias among the included studies.

**Figure 3 jcm-14-04109-f003:**
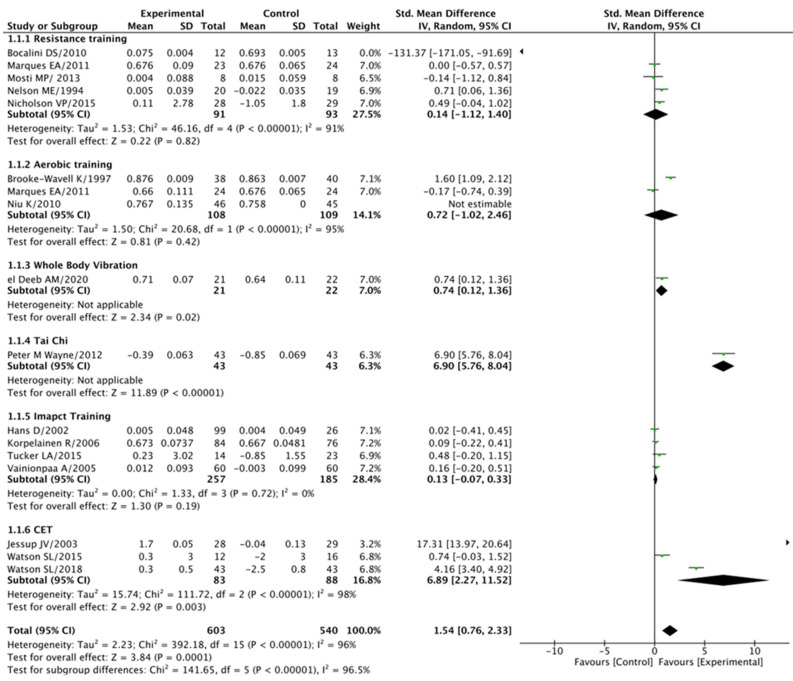
Femoral neck BMD: post-treatment.

**Figure 4 jcm-14-04109-f004:**
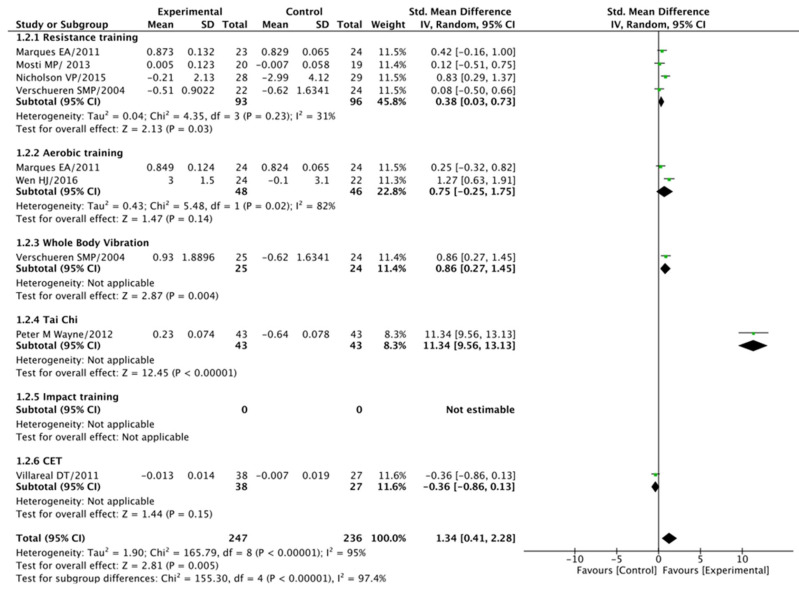
Total hip BMD: post-treatment.

**Figure 5 jcm-14-04109-f005:**
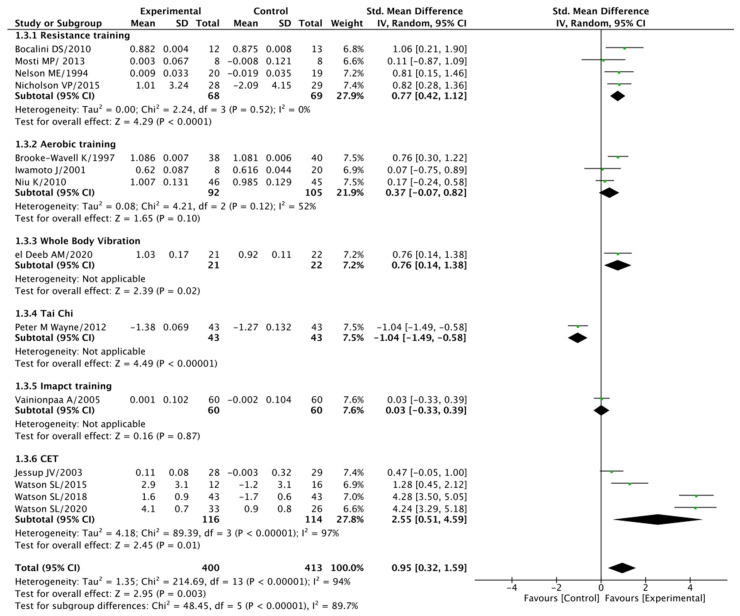
Lumbar spine BMD: post-treatment.

**Table 1 jcm-14-04109-t001:** Summarizes the specific search strategies employed.

No	Databases	Search Terms
1	PEDro	(Osteoporosis OR postmenopausal OR premenopausal OR osteopenia OR Elderly) AND (exercise OR high intensity OR aerobic OR impact training OR high impact or resistance or jumping or weight-bearing exercise) AND (bone mineral density OR bone mass or bone metabolism) AND (quality of life) AND RCT or randomized controlled trial
2	Web of Science	(Osteoporosis OR postmenopausal OR premenopausal OR osteopenia OR Elderly) AND (exercise OR high intensity OR aerobic OR impact training OR high impact or resistance or jumping or weight bearing exercise) AND (bone mineral density OR bone mass or bone metabolism) AND (quality of life)
3	PubMed	Osteoporosis, postmenopausal, premenopausal, osteopenia, or Elderly, in combination with exercise, high intensity, aerobic, impact training, high impact, resistance, jumping, or weight-bearing exercise, as well as bone mineral density, bone mass, or bone metabolism. Additionally, we considered the term quality of life in conjunction with clinical trials, encompassing “clinical trial” [Publication Type], “clinical trials as topic” [MeSH Terms], or “clinical trial” [All Fields].
4	Scopus	TITLE-ABS-KEY Osteoporosis OR postmenopausal OR premenopausal OR osteopenia OR Elderly) AND (exercise OR high intensity OR aerobic OR impact training OR high impact or resistance or jumping or weight bearing exercise) AND (bone mineral density OR bone mass or bone metabolism)
5	EBSCO	Osteoporosis OR postmenopausal OR premenopausal OR osteopenia OR Elderly) AND (exercise OR high intensity OR aerobic OR impact training OR high impact or resistance or jumping or weight bearing exercise) AND (bone mineral density OR bone mass or bone metabolism) AND (“clinical trial)
6	MEDLINE	
7	CINAHL	
8	Science Direct	

**Table 2 jcm-14-04109-t002:** Characteristics of studies on progressive exercise training (PET) for osteoporosis.

S. No	Author/Year	Age (No of Participants)	Participants	Intervention		Exercise Prescription	Outcome Measures	Assessment Intervals	Adverse Effects/Compliance	Site/Equipment Used for BMD Measures	Inference
				EG	CG						
1	Karinkanta et al., 2007 [37]	E: 72.9 ± 2.2 (38) C1: 72.7 ± 2.5 (37) C2: 72.9 ± 2.3 (37) C3: 72.0 ± 2.1 (37)	EW	EG: CET (RT + BT + IT)	CG1: RT CG2: BT + IT CG3: To continue baseline physical activity.	CG1: Intensity-50–60% progressed to 75–80% of 1 RM. Sets: 2 progressed to 3 Repetitions: 10–15 progressed to 8–10 CG2: Gradual escalation in complexity of movements, number of steps, impacts, and jumps.	Assessment of physical performance Self-reported physical functioning Bone mineral content (BMC) Trabecular density measurement Calculation of bone strength index Evaluation of cortical area	Baseline Post 12th month	Falls, ligament injury, fracture, minor knee injuries, partial rupture of muscles and overuse syndrome/In total, 67% of compliance with highest to C1 followed by E and C2 groups	FN, DT, TS, DS, RS/Dual-energy X-ray absorptiometry	Combination of balance, resistance, and impact training improved strength, balance, and BMD.
2	Wayne et al., 2012 [22]	E: 58.8 ± 5.6 (43) C: 60.4 ± 5.3 (43)	PMPW	EG: Tai chi + Usual care	CG: Usual care	CG: Daily consumption of calcium and vitamin D, along with regular exercise as recommended by a healthcare provider. EG: Usual care + 30 min of tai chi exercises, 2 sessions per week for 1st month Later progressed to 3 sessions per week from 2nd to 9th month	BMD BTM SF 36 MENQOL PAR Biomotion sway parameters Clinical balance and function test	Initial assessment After 9 months	Minor musculoskeletal related injuries were reported in 9 participants with E (7) and C (2)/NR	FN, TH, SP/Dual-energy X-ray absorptiometry	Tai chi along with usual care showed better improvements in BMD, QOL, and sway parameters when compared to usual care alone.
3	Iwamoto et al., 2001 [23]	E: 65.3 ± 4.7 (8) C1: 64.3 ± 3.0 (7) C2: 64.9 ± 5.7 (20)	PMPW	EG: AT + Gymnasium	CG1: DT + Gymnasium CG2: No exercise	Experimental Group (EG): Participants were motivated to enhance their daily step count through brisk walking for a duration of 2 years, combined with two sets of gymnastic training sessions (including exercises like SLR, squatting, abdominal, and back strengthening exercises) for 2 years. Control Group 1 (CG1): Participants were encouraged to increase their daily step count by brisk walking for 1 year, along with two sets of gymnastic training sessions (comprising exercises such as SLR, squatting, abdominal, and back strengthening exercises) conducted 5 days per week for a duration of 2 years. Control Group 2 (CG2): Participants did not engage in any specific exercise regimen. All participants received calcium lactate and vitamin D supplementation for a period of 2 years.	- (BMD) - Serum calcium levels - Serum phosphorus levels - Serum alkaline phosphatase levels - Daily step count	- Baseline - 1-tear follow-up - 2-year follow-up	Not reported	LS/Dual-energy X-ray absorptiometry	Sustained exercise training is necessary to preserve the bone mass acquired through exercise training.
4	Brooke-Wavell et al., 2001 [24]	E: 64.9 ± 3.0 (38) C: 64.2 ± 3.1 (40)	PMPW	EG: AT	CG: Swimming routine physical activity	EG: Progressively increased the duration from 120 min to 280 min. Frequency: 20–50 min per day for 12 months CG: 20 min of swimming per session 2 sessions per week for 12 months.	Bone mineral density Body mass Body mass index Broadband Ultrasound Attenuation (BUA) Maximum oxygen uptake	Baseline Post 12 months	Two women reported minor walking-related foot injuries	Lumbar spine (LS), Femoral neck (FN), and calcaneum assessments were conducted utilizing Dual-Energy X-ray Absorptiometry (DXA).	Walking mitigated bone loss, particularly in the calcaneus, and may have had a positive impact on bone density in the lumbar spine. Additionally, it improved functional capacity and thwarted the increase in body mass that was noted in the control group.
5	Korpelainen et al., 2006 [25]	E: 72.9 ± 1.1 (84) C: 72.8 ± 1.2 (76)	EW	EG: IT	CG: Continue with regular activities	EG: The supervised and home exercise regimens were regularly adjusted every two months to ensure continued progress and variety. Participants engaged in 1 h of supervised impact exercises along with a 20 min home exercise program for a duration of 30 months.	BMD BMC	Initial assessment Assessment after 12 months Assessment after 30 months	Three female participants reported minor injuries. In the experimental group, six individuals experienced fractures, while in the control group, 16 individuals had fractures. The compliance rate for supervised sessions ranged from 73% to 78%	TF, TR, and FN/Peripheral DXA (Osteometer DTX 200, Osteometer Meditech, Roedovre, Denmark.)	Impact exercise did not significantly influence (BMD), but it did have a positive impact on bone mineral content (BMC) at the trochanter. This exercise regimen may help reduce the risk of fractures related to falls in elderly women with low bone mass.
6	Hans et al., 2002 [26]	E: 67.6 ± 5.2 (99) C1: 66.3 ± 7.6 (32) C2: 66.0 ± 4.8 (26)	PMPW	EG: IT	CG1: Heel drops without impact. CG2: Continue with regular activities.	In the experimental group (EG), the exercise parameters were as follows: - Amplitude: 25 ± 50% above the patient’s estimated resting force - Rise time: Less than 20 milliseconds - Repetition rate: 1/3 ± 2/3 Hertz - Repetitions: 120 correct force impacts daily at home, completed in one 3–5-min period - Progression: The force range was initially set at 25% above resting force and gradually increased to 50% above normal gravitational force.	BMD	- Initial assessment - Evaluation after 12 months - Assessment after 18 months - Final evaluation after 24 months	Slight backache without any serious adverse events/91% and 65% of compliance at 18 months and 24 months, respectively	FN, IT, TR, Ward’s tr/Hologic QDR 1000 or 2000 densitometers	Maintaining hip BMD could potentially be achieved with a short, secure, supervised impact loading program conducted at home.
7	Verschueren et al., 2004 [27]	E: 64.6 ± 3.3 (25) C1: 63.90 ± 3.8 (22) C2: 64.2 ± 3.1 (24)	PMPW	EG: WBV	CG1: Resistance training. CG2: Continue with regular activities.	Experimental Group (EG) Protocol: - Vibration Intensity: Varied from 1.7 mm to 2.5 mm - Frequency: Ranged between 35–40 Hz - Duration per session: 30 min - Progression involved reducing rest periods, increasing training load, and transitioning from two-legged to one-legged exercises. Control Group 1 (CG1) Protocol: - Exercise Intensity: Maintained within the range of 60% to 80% of the heart rate reserve. - Duration per session: 60 min. - Progression included transitioning from a 20-repetition maximum (RM) to two sets of 15 RM, two sets of 12 RM, two sets of 10 RM, and eventually two sets of 8 RM. - In the final 10 weeks, training volume and intensity ranged from three sets of 12 RM to one set of 8 RM. - Each resistance exercise program had a typical duration of approximately 1 h.	(BMD) Isometric muscular strength Skeletal muscle volume Adipose tissue volume Postural stability	Initial measurement After a half-year period	Not reported	Measurement of trabecular thickness (TH) and total bone density (TB) through Dual-Energy X-ray Absorptiometry (DXA) utilizing the QDR-4500A apparatus (Hologic, Waltham, MA, USA).	Whole Body Vibration (WBV) training appears to be a viable and efficacious approach for altering established risk factors associated with falls and fractures in elderly women, underscoring the necessity for additional human research in this area.
8	Nicholson et al., 2015 [28]	E: 66.0 ± 4.1 (28) C: 65.6± 4.7 (29)	PMPW	EG: RT	CG: Continue with regular activities.	EG: 50 min of 2 sessions per week for 6 months Progression: Systematic increase in the load	(BMD) Body mass One-Repetition Maximum (1 RM) Physical energy Dietary protein Calcium intake Metabolic Equivalents (METs)	Initial assessment After a half-year period	One participant complaint of exacerbation of knee pain/89% compliance	Lumbar Spine (LS), Femoral Neck (FN), Total Hip (TH), Total Radius (TR), and Total Body (TB) assessments were conducted using Pencil Beam Dual-Energy X-ray Absorptiometry (DXA) technology, specifically the Lunar DPX Pro system from GE Healthcare based in the U.K.	Low-load, high-repetition resistance training has proven effective in mitigating declines in lumbar spine (BMD) when compared to control groups in healthy and physically active women aged over 55 years. However, this type of training did not exhibit a significant impact on BMD in the hip and total body, nor did it influence measures of fat mass and fat-free soft tissue mass.
9	Jessup et al., 2003 [29]	E: 69.1 ± 2.8 (28) C: 69.4 ± 4.2 (29)	PMPW	EG: CET (RT + BT + AT)	CG: No exercise	EG: For instance: Duration: 60–90 min per session, Frequency: Three sessions per week for a total of 32 weeks. Progression: RT: 8–10 repetitions with 50% of 1 RM progressed to 75% of 1 RM. BT + AT: Started with no weights progressed to 10% of body weight with increase in complexity of exercise.	BMD Body sway Strength Osteoporosis self-efficacy scale Body Weight	Initial measurement After 32 weeks	No adverse effects were reported/Not reported	Femoral neck and Lumbar spine measurements were obtained using the Norland Excel DEXA scan system manufactured by Norland Medical Systems in White Plains, New York.	The experimental group (EG) experienced substantial enhancements in femoral neck bone density and balance, coupled with notable weight loss. Self-efficacy levels remained unchanged in both groups.
10	Paolucci et al., 2014 [30]	E: 65.6 ±5.8 (40) C: 65.6 ±5.3 (20)	PMPW	EG: Supervised CET (RT + AT + BT)	CG: Unsupervised MM	Both the Experimental Group (EG) and Control Group (CG) engaged in low-intensity exercise sessions lasting 60 min each. These sessions occurred three times per week and spanned a total of 10 sessions. The exercise programs included a progression in exercise intensity over time.	VAS MPQ ODQ SOQLQ	Initial assessment After 10 sessions Follow-up at 6 months	The attendance rates for the Experimental Group (EG) sessions were not reported. However, it is noteworthy that a high attendance rate was observed, with 93% of participants attending all sessions.	-	Supervised multimodal exercises have demonstrated their effectiveness in reducing back pain and enhancing functional status and quality of life among women coping with postmenopausal osteoporosis. Importantly, these positive outcomes have been sustained for a period of six months.
11	Watson et al., 2018 [32]	E: 65 ± 5 (43) C: 65 ± 5 (43)	PMPW	EG: CET (RT + IT)	CG: RT	EG and CG: Duration: 30 min per session, Frequency: 2 times per week for 8 months Progression: EG: RT: Intensity progressed from 50–70% to 80–85% of 1 RM and IT: jumping with flexed lower limb position to stiff leg landing. C: Progressively adding weights up to 3 kg	BMD T-score BES TUGT FTSTS FRT cBPAQ Daily calcium intake	Baseline Post 8 months	One participant had mild low back pain in the experimental group/92 ± 11% for the experimental group and 85± 24%, for the control group.	LS, and FN/Dual-energy X-ray absorptiometry	Short-term, supervised multimodal exercise programs have shown the potential to improve BMD and physical performance in postmenopausal women who have low to very low bone mass.
12	Watson et al., 2015 [31]	E: 65.3 ± 3.9 (12) C: 66.7 ± 5.4 (16)	PMPW	EG: CET (RT + IT)	CG: RT	EG and CG: Duration: 30 min per session, Frequency: 2 times per week for 8 months Progression: E: RT: Intensity progressed from 50–70% to 80–85% of 1 RM and IT: jumping with flexed lower limb position to stiff leg landing C: Progressively adding weights up to 3 kg	BMD T-score BES TUGT FTSTS FRT	Baseline Post 8 months	No adverse events have been reported/87.2 ± 3.9% for the experimental group and 92.7 ± 3.8%, for the control group.	LS, and FN/Dual-energy X-ray absorptiometry	Brief supervised multimodal exercise is considered a safe and effective exercise therapy for postmenopausal women who have low to very low bone mass.
13	Harding et al., 2020 [41]	E: 64.9 ± 8.6 (33) C: 67.4 ± 6.3 (26)	Middle age and older men	EG: CET (RT + IT)	CG: Continue with regular activities.	EG and CG: Duration: 30 min per session, Frequency: 2 times per week for 8 months Progression: EG: RT: Intensity progressed from 50–70% to 80–85% of 1 RM with RPE ≥16 on the 6-to-20-point Borg scale and IT: jumping with flexed lower limb position to stiff leg landing	BMD T-score cBPAQ Daily calcium intake Body fat percentage	Baseline Post 8 months	Five mild musculoskeletal discomfort occurred and muscle soreness/77.8% 16.6% for experimental group and 78.5% 14.8% for the control group.	LS, FN, TH and TR/Dual-energy X-ray absorptiometry	Multimodal exercise improved BMD, function and facture risk when compared to control group.
14	Teixeira et al., 2010 [33]	E: 63.1± 4.5 (33) C: 62.7 ± 4.8 (26)	PMPW	EG: CET (RT + PT + +BT+ DT)	CG: DT	Experimental Group (EG) Protocol: - Frequency: Twice a week for a duration of 18 weeks. - Progression in the Physical Therapy and Balance Training (PT + BT) included: - Transition from stable surfaces to unstable surfaces. - Advancement from gait training without obstacles to gait training with obstacles. - Commencement of exercises with eyes open and later with eyes closed. - Starting with low-speed exercises and, based on patient performance, progressing to high-speed exercises. - Initially, bipedal training, followed by unipedal training. - Utilization of resources such as balance and trampoline. - Resistance Training (RT) progression involved: - Intensity increment from 50% to up to 80% of the 1-Repetition Maximum (1-RM).	SF-36 BBS TUG DST	Baseline Post 18 Weeks Post 6 months follow up	Five and six participants reported falls after treatment sessions in experimental and control groups, respectively/82 ± 5.83% of compliance rate.	-	The combination of progressive strength training for the quadriceps and proprioceptive training has proven to be effective in preventing falls. This approach enhances muscle power and both static and dynamic balance, and accelerates motor response times, ultimately leading to improved performance in daily activities.
15	Villareal et al., 2011 [38]	E: 70± 4 (38) C1: 70 ± 4 (26) C2: 70 ± 4 (26) C3: 69 ± 4 (27)	EW	EG: CET (RT + AT + Diet)	CG1: RT+ AT CG2: Diet CG3: No diet and exercises	Both the Experimental Group (EG) and Control Group 1 (CG1) engaged in exercise sessions with the following parameters: - Duration: 90 min - Frequency: Three sessions per week for a period of 12 months Progression in exercise intensity for each group was as follows: Aerobic Training (AT): - Initially, participants exercised at an intensity of 65% of their peak heart rate. - Over time, they gradually increased the intensity to a range of 70% to 85% of their peak heart rate. - Resistance Training (RT): - At the outset, participants performed 1 or 2 sets of exercises at a resistance level of approximately 65% of their one-repetition maximum (1-RM). - As the program advanced, they gradually intensified the resistance to 2 to 3 sets of exercises, with the resistance set at approximately 80% of their one-repetition maximum. - Repetitions for each exercise ranged from 8 to 12 initially and reduced to 6 to 8 repetitions as the program progressed.	BMD Physical performance test and other measures of frailty Body weight and composition Muscle strength Balance Gait SF-36	Baseline Post 6 months Post 1 year	During the study, one participant in the research cohort experienced an ankle fracture. The compliance rates for the interventions were as follows: - In the exercise group, participant compliance was 88%. - In the diet + exercise group, participant compliance was 83%. - In the diet-only group, participant compliance was also 83%.	TH/Dual-energy X-ray absorptiometry	The combination of weight loss and exercise has been found to yield greater improvements in physical function when compared to either intervention alone.
16	Wen et al., 2017 [34]	E: 57.5 ± 3.5 (24) C: 58.8 ± 3.2 (22)	PMPW	EG: AT	CG: No exercises	Experimental Group (EG) Protocol: - Duration: 1 h and 30 min per session - Frequency: Three times per week for a total duration of 10 weeks - Intensity: Maintained between 75% to 85% of the heart rate reserve - Progression in the exercise regimen included an increase in step height over the course of the program.	Body composition Lipid profile Functional fitness BMC BMD BA Bone turnover Strength Balance Dietary intake	Baseline Post 10 weeks	No adverse events occurred/96.7 ± 0.9% compliance towards the experimental group.	TH and TB/Dual-energy X-ray absorptiometer	Short-term step aerobic exercise showed significant improvements on bone metabolism and general health but not on BMD.
17	Niu et al., 2010 [21]	E: 38.1 ± 1.2 (46) C: 39.7 ± 1.2 (45)	PRMPW	EG: AT	CG: SE	EG and CG: Duration:16 min, Frequency: Three times per week for 12 months. Intensity: 5 X10 vertical and versatile jumps in experimental group. Progression: EG: Progressed up to 50 jumps by 3 months; 6 months onwards, jumping from 10 cm step.	BMD	Baseline Post 12 months	No adverse events occurred/2.4 (0.8–3.2) times per week for both groups.	LS, PF, FN, IT, TF, Ward’s triangle/Dual-energy X-ray absorptiometer	The experimental group demonstrated statistically significant changes in Femoral Neck Bone Mineral Density (BMD) when compared to the control group.
18	Tucker et al., 2015 [42]	E1: 41.09 ± 4,3(23) E2: 39.79 ± 4.7(14) C: 37.65± 6.4 (23)	PRMPW	EG1: IT (10 jumps) EG2: IT (20 jumps)	CG: SE	Participants were instructed to perform either 10 or 20 jumps during each session, with a total of 2 sessions per day. This regimen was followed for 6 days per week over a span of 16 weeks.	BMD Anthropometric measurements Calcium intake	Baseline Post 8 weeks Post 16 weeks	Adverse events not reported/73% compliance	TH/Dual-energy X-ray absorptiometry	Hip BMD improved in both jumping groups compared to control group.
19	Vainionpää et al., 2005 [43]	E: 38.1± 1.7 (60) C: 38.5 ± 1.6 (60)	PRMPW	EG: IT	CG: No exercise	Experimental Group (EG) Protocol: - Duration: 60 min per session - Frequency: Three sessions per week for a duration of 12 months - Progression within the exercise regimen included: - After 3 months: An increase in the height of the bench by one step (with a height of 10 cm). - After 6 months: Further advancement with an increase in bench height by two or three step benches.	BMD	Baseline Post 12 months	No adverse events occurred/Not reported	FN, TR, IT, TF, Ward’s triangle, LS, RA, UL, DR, CL/Dual-energy X-ray absorptiometry	High-impact exercises have been shown to lead to improvements in BMD, specifically in the lumbar spine and upper femur among premenopausal women.
20	Nelson et al., 1994 [35]	E: 61.1 ± 3.7 (20) C: 57.3 ± 6.3 (19)	PMPW	EG: RT	CG: Continue with regular activities.	EG: Duration: 45 min, Frequency: 2 days per week for 54-week, Intensity: 50% and 60% of the baseline 1 RM with 16 on the Borg scale Progressed to 80% of 1 RM.		Initial assessment After 24 weeks	Seven participants in the exercise group complained of mild musculoskeletal pain. One woman suffered an ankle sprain and two others suffered wrist fractures due to falls in the control group/87.5 ± 1.8% compliance to the experimental group.	LS, FN/Dual-energy X-ray absorptiometry	High-intensity strength training retains BMD and improves muscle mass, strength, and balance.
21	Mosti et al., 2013 [39]	E: 61.9 ± 5.0 (8) C: 66.7 ± 7.4 (8)	EW	EG: RT	CG: Continue with osteopenia exercise guidelines.	Experimental Group (EG) followed a regimen consisting of three sessions per week over a 12-week period. The intensity level involved four sets, each comprising 3–5 repetitions, with a resistance set at 85–90% of their initial 1 RM (one-repetition maximum). If participants successfully completed 5 repetitions, the training load was elevated by 2.5 kg to ensure progression.	(BMD) Bone Mineral Content (BMC) Serum markers of bone health	Baseline Post 54 weeks	No adverse events occurred/87% compliance to exercise program.	LS, FN, and TH/Dual X-ray Absorptiometry	Maximum strength training program improves BMD.
22	Bocalini et al., 2010 [36]	E: 66 ± 9 (12) C: 64 ± 8 (13)	PMPW	EG: RT	CG: No exercises	The Experimental Group (EG) engaged in 60-min sessions, three times a week, over a 24-week period. The initial intensity was set at 40% of their 1 RM (one-repetition maximum). Progression involved performing three sets of 10–12 repetitions for the specific exercise at an intensity level of 60–70% of their 1 RM.	(BMD) Body weight Body Mass Index (BMI) Body fat percentage Lean body mass Maximal aerobic capacity	Initial assessment After 54 weeks	No adverse events occurred/Not reported.	LS, FN,/Dual X-ray Absorptiometry	RT suppresses the decline in BMD and simultaneously improves the functional fitness of postmenopausal women.
23	ElDeeb and Abdel-Aziem, 2020 [44]	E: 55.09 ± 4.19 (21) C: 57.29 ± 4.44 (22)	PMPW	EG: WBV	CG: CG	Duration: 5–10 min, Frequency: 2 sessions per week for 24 weeks Intensity: Frequency 20 Hz, amplitude ranged from 2.5 to 5 mm, 5 min, holding the position for 30 s, rest period 45 s with 3 repetitions. Progression: Progression in frequency up to 35 Hz, amplitude, position holding time up to 60 s, rest period reduced to 5 s and repetitions up to 9 by 6 months.	Muscle work BMD	• Initial assessment - After 24 weeks		LS, FN, Ward’s triangle, and GT/Dual X-ray Absorptiometry	Whole body vibration improved muscle work and BMD.
24	Marques et al., 2011 [40]	E1: 67.3 ± 5.2 (23) E2: 70.3 ± 5.5 (24) C: 67.9 ± 5.9 (24)	EW	EG1: RT EG2:AT	CG: No exercises	EG1: 60 min per session, 3 sessions per week or 32 weeks. Progression: 50–60% of 1 RM, 2 sets of 10–15 repetitions progressed to 75–80% of 1 RM, 2 sets 6–8 repetitions. EG2: 60 min per session, 3 sessions per week or 32 weeks Progression: Exercise intensity at initial weeks was 50–60% of heart reserve later progressed to 65% to 80% of heart rate reserve.	BMD	Initial measurement After 32 weeks	No adverse events related to exercise or assessments were reported during the study. The compliance rate for resistance exercise (RE) sessions was 78.4%, with a range from 61.6% to 95.9%. For aerobic exercise (AE) training, the mean compliance rate was 77.7%, with a range from 64.2% to 96.8%.	FN, TR, IT, and TH/Dual X-ray Absorptiometry	

**Table 3 jcm-14-04109-t003:** Quality assessment for randomized control trials (RCTs) using the Physiotherapy Evidence Database (PEDro) scale.

S. No.	Author/References	Eligibility Criteria	Random Allocation	Concealed Allocation	Baseline Comparability	Blinding of Participants	Blinding of Therapist	Blinding of Assessor	Adequate Follow Up (>85%)	Intention to Treat Analysis	Between Group Comparison	Point Estimates and Variability	PEDro Score (10)
1	Karinkanta et al., 2007 [37]	Yes	Yes	Yes	Yes	No	No	No	Yes	Yes	Yes	Yes	7
2	Wayne et al., 2012 [22]	No	Yes	No	Yes	No	No	Yes	Yes	Yes	Yes	Yes	7
3	Iwamoto et al., 2001 [23]	No	Yes	No	Yes	No	No	No	No	No	Yes	Yes	4
4	Brooke-Wavell et al., 2001 [24]	Yes	Yes	No	Yes	No	No	Yes	Yes	No	Yes	Yes	6
5	Korpelainen et al., 2006 [25]	Yes	Yes	No	Yes	No	No	Yes	No	Yes	Yes	Yes	6
6	Hans et al., 2002 [26]	No	Yes	Yes	Yes	No	No	No	No	No	Yes	Yes	5
7	Verschueren et al., 2004 [27]	No	Yes	No	Yes	No	No	Yes	No	No	Yes	Yes	5
8	Nicholson et al., 2015 [28]	Yes	Yes	No	Yes	No	No	Yes	Yes	Yes	Yes	Yes	7
9	Jessup et al., 2003 [29]	Yes	Yes	No	Yes	Yes	Yes	Yes	Yes	No	Yes	Yes	8
10	Paolucci et al., 2014 [30]	Yes	Yes	No	Yes	Yes	No	No	No	No	Yes	Yes	5
11	Watson et al., 2015 [31]	Yes	Yes	No	Yes	Yes	No	No	No	No	Yes	Yes	5
12	Watson et al., 2018 [32]	Yes	Yes	Yes	Yes	No	No	Yes	Yes	Yes	Yes	Yes	8
13	Harding et al., 2020 [41]	Yes	Yes	Yes	Yes	Yes	No	Yes	Yes	Yes	Yes	Yes	9
14	Teixeira et al., 2010 [33]	Yes	Yes	Yes	Yes	No	No	Yes	No	No	Yes	Yes	6
15	Villareal et al., 2011 [38]	Yes	Yes	No	Yes	No	No	Yes	Yes	Yes	Yes	Yes	7
16	Wen et al., 2017 [34]	Yes	Yes	No	Yes	No	No	No	Yes	No	Yes	Yes	5
17	Niu et al., 2010 [21]	Yes	Yes	No	Yes	No	No	No	No	No	Yes	Yes	4
18	Tucker et al., 2015 [42]	Yes	Yes	No	Yes	No	No	No	No	No	Yes	Yes	4
19	Vainionpää et al., 2005 [43]	Yes	Yes	No	Yes	No	No	No	No	No	Yes	Yes	4
20	Nelson et al., 1994 [35]	Yes	Yes	No	No	No	No	No	Yes	Yes	Yes	Yes	5
21	Mosti et al., 2013 [39]	Yes	Yes	No	Yes	No	No	No	No	No	Yes	Yes	4
22	Bocalini et al., 2010 [36]	Yes	Yes	No	Yes	No	Yes	No	No	No	Yes	Yes	5
23	ElDeeb and Abdel-Aziem, 2020 [44]	No	Yes	Yes	Yes	No	No	No	Yes	No	Yes	Yes	6
24	Marques et al., 2011 [40]	Yes	Yes	Yes	Yes	No	No	Yes	No	Yes	Yes	Yes	7

## Data Availability

All data is provided via tables in the text. The included articles are available via PubMed.

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
