# Peer review of "Effectiveness of Exercise Loading on Bone Mineral Density and Quality of Life Among People Diagnosed with Osteoporosis, Osteopenia, and at Risk of Osteoporosis—A Systematic Review and Meta-Analysis"

_jcm, 2025, doi:10.3390/jcm14124109_

Round 1

Reviewer 1 Report

Comments and Suggestions for Authors

This systematic review/meta-analysis evaluates the efficacy of Progressive Exercise Training (PET) on bone mineral density (BMD) and quality of life (QOL) in individuals with osteoporosis, osteopenia, or at risk of osteoporosis.

Note in the upper left corner the type of paper according to the journal template.

The abstract is structured; the keywords should be checked in accordance with MeSH.

The introduction should better summarize the burden and health risks of osteoporosis, in relation to scientific literature (for e.g. doi: 10.3390/jcm14093162 ). The objectives should be stated as (RQ) at the end of this section

For the methodology and results section: Pleas provide the manufacturer and country of the “Review Manager 5.4.1” software. While registration DOI is mentioned, the protocol’s details (e.g., inclusion criteria, outcomes) are not cross-referenced. Use dots between the initials – line 158 “(SMA & SN).” Table 1 lacks specificity, making replication difficult; for example, terms like "high impact" or "resistance training" should be standardized across databases. I² values up to 98% (e.g., combined exercise training) suggest variability in protocols or populations that need clarification; please add a sensitivity analyses or meta-regression to explore sources of heterogeneity (e.g., exercise intensity, participant age, baseline BMD).

In the discussion section please emphasize more recent advances of progressive exercise training (PET) taking examples from Ciobanu et al. pilot study of a mechatronic system for gait rehabilitation. Future research directions should be better contoured at the end of this section given the type of paper.

Editing recommendations: 1. The references should be noted with “[ ]” rather than in superscript; Please edit Table 2 in APA academic style also. 3. Number the sections and subsections.

Some of the references are outdated (for e.g. [5] - 1999) and should be exchanged with newer ones as suggested above.

Author Response

Reviewer 1

We want to thank the reviewers for providing us with an opportunity to rewrite the manuscript. Their comments and constructive suggestions helped us to improve this manuscript's quality. The reviewer's comments and authors' responses to the comments are given below

Reviewer comment 1: Note in the upper left corner the type of paper according to the journal template.

Authors' response to reviewer's comment : Corrected as per the suggestion

Reviewer comment 2:

The abstract is structured; the keywords should be checked in accordance with MeSH

Authors' response to reviewer's comment : Thank you for the comments . All the five keywords used are valid MeSH terms

Reviewer comment 3:

The introduction should better summarize the burden and health risks of osteoporosis, in relation to scientific literature (for e.g. doi: 10.3390/jcm14093162 )

Authors' response to reviewer's comment : Accordingly, we have revised the section to provide a more comprehensive summary of the global prevalence, consequences, and public health impact of osteoporosis. In addition to incorporating data from the suggested reference (DOI: 10.3390/jcm14093162), we have also included findings from an additional relevant study to further support the rationale for our research.

Reviewer comment 4:

The objectives should be stated as (RQ) at the end of this section

Authors' response to reviewer's comment : Thank you for your suggestion. We have revised the final part of the Introduction to explicitly state the research objective in the form of a research question (RQ), as recommended.

Reviewer comment 5: For the methodology and results section: Pleas provide the manufacturer and country of the “Review Manager 5.4.1” software.

Authors' response to reviewer's comment: Thank you for your suggestion. We have now added the manufacturer and country information for the Review Manager software

Reviewer comment 6: While registration DOI is mentioned, the protocol’s details (e.g., inclusion criteria, outcomes) are not cross-referenced.???

Authors' response to reviewer's comment: Thank you for the observation. We would like to clarify that the eligibility criteria is already mentioned in the methodology section of the manuscript.

Reviewer comment 7: Use dots between the initials – line 158 “(S.M.A & S.N).”

Authors' response to reviewer's comment: Corrected as per the suggestion

Reviewer comment 8: Table 1 lacks specificity, making replication difficult; for example, terms like "high impact" or "resistance training" should be standardized across databases.

Authors' response to reviewer's comment : We thank the reviewer for this suggestion. We acknowledge that using standardized and database-specific terminology would improve the clarity and reproducibility of the search strategy. However, as the literature search has already been completed and the manuscript is now prepared for publication, it is not possible to revise the search strategy retrospectively. We greatly appreciate your understanding and will ensure more standardized reporting in future systematic reviews.

Reviewer comment 9: I² values up to 98% (e.g., combined exercise training) suggest variability in protocols or populations that need clarification; please add a sensitivity analyses or meta-regression to explore sources of heterogeneity (e.g., exercise intensity, participant age, baseline BMD).

Authors' response to reviewer's comment : We thank the reviewer for this suggestion. We acknowledge the substantial heterogeneity observed in some pooled results. To address this, we performed subgroup analyses based on the type of exercise intervention, which helped partially explain the variability. However, sensitivity analysis or meta-regression was not conducted due to the limited number of studies per subgroup and inconsistent reporting of potential moderators such as exercise intensity, participant age, or baseline BMD. We have now clarified this in the limitations section of the manuscript as

‘although subgroup analysis has been conducted in our study based on the type of exercises to find out potential source of heterogeneity ,substantial  variability is no-ticed in some of the pooled outcomes. However sensitivity  analyses or meta regres-sion were not performed due to limited number of researches within subgroup and inconsistent reporting of variables such as exercise intensity , age and bone mineral density .This restrict our ability to further investigate the source of heterogeneity and it should be addressed in future studies with more comprehensive data’.

Reviewer comment 10: In the discussion section please emphasize more recent advances of progressive exercise training (PET) taking examples from Ciobanu et al. pilot study of a mechatronic system for gait rehabilitation. Future research directions should be better contoured at the end of this section given the type of paper.

Authors' response to reviewer's comment : The above-mentioned device is designed for stroke patients who can’t stand upright in a dependent position. Our study population is people with Osteoporosis, Osteopenia, and at Risk of Osteoporosis, all of whom are supposed to be able to assume the upright standing and perform exercises suitable to their condition. We explored the literature for similar types of devices that can apply certain loads to bone to simulate its growth. For example, Prabhala et al. (2016) proposed a knee loading device to stimulate bone growth. However, not enough studies were found to support the use of this device. They had only one citation since 2016. With regard to the second half of the comment concerning the future research direction, it is discussed at the end of the discussion section under the subtitle “Implications and Future Directions”

Reviewer comment 11: Editing recommendations: 1. The references should be noted with “[ ]” rather than in superscript;

Authors' response to reviewer's comment : Corrected as per the suggestion

Reviewer comment 12: Please edit Table 2 in APA academic style also. 3.

Authors' response to reviewer's comment : Table 2 updated as per the suggestion

Reviewer comment 13: Number the sections and subsections.

Authors' response to reviewer's comment : updated as per the suggestion

Reviewer comment 14: Some of the references are outdated (for e.g. [5] - 1999) and should be exchanged with newer ones as suggested above.

Authors' response to reviewer's comment : Some of the outdated references updated

Reviewer 2 Report

Comments and Suggestions for Authors

Saeed Mufleh Alnasser et al. comprehensively assessed the effects of PET on BMD and quality of life through systematic review and meta-analysis, especially for those with osteoporosis, decreased bone mass, or at risk of osteoporosis, to inform the management of osteoporosis. The article is logically coherent and clearly written, but there are still some areas for improvement. Here are my recommendations:

1) There are few studies on the assessment of quality of life, and there is a lack of in-depth discussion on PET improving quality of life;

2) There is a high risk of bias in the study, such as inadequate control in random allocation and blinding design, which may affect the reliability of the results;

3) The heterogeneity of the results is large, especially in the effects of different types of exercise interventions, which may affect the general applicability of the results;

4) It is important to note that this article has a 24% check rate. The authors should revise it carefully to reduce the repetition rate to less than 20%.

Author Response

Reviewer 2

We want to thank the reviewers for providing us with an opportunity to rewrite the manuscript. Their comments and constructive suggestions helped us to improve this manuscript's quality. The reviewer's comments and authors' responses to the comments are given below

:

Reviewer comment 1:  There are few studies on the assessment of quality of life, and there is a lack of in-depth discussion on PET improving quality of life; ???

Authors' response to reviewer's comment : We agree that the assessment and discussion of quality of life (QoL) are important aspects of this topic. In response, we have addressed this in detail in Section 4.4, where two dedicated paragraphs discuss the impact of physical exercise therapy (PET) on quality of life, supported by relevant literature.

Reviewer comment 2: There is a high risk of bias in the study, such as inadequate control in random allocation and blinding design, which may affect the reliability of the results;

Authors' response to reviewer's comment : The authors also  agree with the reviewer’s observation regarding the potential risk of bias due to limitations in random allocation and blinding. This point has been acknowledged and added to the Limitations section of the manuscript as

‘Second, there were an inadequate control in random allocation and blinding design of the included studies which may have increased the risk of bias and therefor the reliability of results. This may highlight the need for more studies with better methodological quality allowing the current systematic review to be repeated in the future.’

Reviewer comment 3: The heterogeneity of the results is large, especially in the effects of different types of exercise interventions, which may affect the general applicability of the results;

Authors' response to reviewer's comment: Thank you for the comments . This has been acknowledged in the limitation section of the manuscript 

Reviewer comment 4:  It is important to note that this article has a 24% check rate. The authors should revise it carefully to reduce the repetition rate to less than 20%.

Authors' response to reviewer's comment : The authors reduced  the plagiarism during revision

Round 2

Reviewer 1 Report

Comments and Suggestions for Authors

The authors have improved their paper accordingly. 

Reviewer 2 Report

Comments and Suggestions for Authors

The author has made revisions to the questions I raised earlier. It has been sufficiently improved to warrant publication in JCM.